# Identifying Key Genes as Progression Indicators of Prostate Cancer with Castration Resistance Based on Dynamic Network Biomarker Algorithm and Weighted Gene Correlation Network Analysis

**DOI:** 10.3390/biomedicines12092157

**Published:** 2024-09-23

**Authors:** Siyuan Liu, Yi Hu, Fei Liu, Yizheng Jiang, Hongrui Wang, Xusheng Wu, Dehua Hu

**Affiliations:** 1School of Life Sciences, Central South University, Changsha 410013, China; 212501030@csu.edu.cn (S.L.); 15243664769@163.com (Y.H.); jyzgenetic@outlook.com (Y.J.); 223254@csu.edu.cn (H.W.); 2Shenzhen Health Development Research and Data Management Center, Shenzhen 518028, China; liufei_csu@163.com

**Keywords:** castrate-resistant prostate cancer, dynamic network biomarker, risk prediction model, bulk RNA sequence, single-cell RNA sequence

## Abstract

**Background:** Androgen deprivation therapy (ADT) is the mainstay of treatment for prostate cancer, yet dynamic molecular changes from hormone-sensitive to castration-resistant states in patients treated with ADT remain unclear. **Methods:** In this study, we combined the dynamic network biomarker (DNB) method and the weighted gene co-expression network analysis (WGCNA) to identify key genes associated with the progression to a castration-resistant state in prostate cancer via the integration of single-cell and bulk RNA sequencing data. Based on the gene expression profiles of CRPC in the GEO dataset, the DNB method was used to clarify the condition of epithelial cells and find out the most significant transition signal DNB modules and genes included. Then, we calculated gene modules associated with the clinical phenotype stage based on the WGCNA. IHC was conducted to validate the expression of the key genes in CRPC and primary PCa patients Results:Nomograms, calibration plots, and ROC curves were applied to evaluate the good prognostic accuracy of the risk prediction model. **Results:** By combining single-cell RNA sequence data and bulk RNA sequence data, we identified a set of DNBs, whose roles involved in androgen-associated activities indicated the signals of a prostate cancer cell transition from an androgen-dependent state to a castration-resistant state. In addition, a risk prediction model including the risk score of four key genes (SCD, NARS2, ALDH1A1, and NFXL1) and other clinical–pathological characteristics was constructed and verified to be able to reasonably predict the prognosis of patients receiving ADT. **Conclusions:** In summary, four key genes from DNBs were identified as potential diagnostic markers for patients treated with ADT and a risk score-based nomogram will facilitate precise prognosis prediction and individualized therapeutic interventions of CRPC.

## 1. Introduction

Prostate cancer (PCa) is the most prevalent cancer for men in the USA and a leading cause of oncological death in men worldwide, with an estimated 35,000 deaths and more than 299,000 new cases in the US in 2024 [1]. For most PCa patients, castration-resistant prostate cancer (CRPC), whether metastatic (mCRPC) or nonmetastatic (nmCRPC), generally occurs in response to therapeutic pressure, specifically the use of androgen deprivation therapy (ADT) [2]. Worse still, almost all patients treated with ADT eventually develop castration-resistant prostate cancer (CRPC), as evidenced by imaging progression or an increased prostate-specific antigen (PSA) despite castration levels of testosterone [3]. CRPC has a poor prognosis with an average survival time of only 16–18 months from progression [4], while the overall survival of patients with metastatic CRPC is only 9–13 months [5]. Currently, the prognosis and treatment of CRPC remain major challenges and the exact mechanism of the transition from a hormone-sensitive to a castration-resistant state is still not fully understood [6,7], Thus, there is a critical need to explore the molecular markers between hormone-naive prostate cancer and CRPC to better understand the mechanism by which primary prostate cancer transforms to CRPC and provide new targets for treating CRPC.

There have been a range of studies concerning progression indicators and drug resistance biomarkers of CRPC. For instance, glutamate decarboxylase 1 (GAD1) was found to promote prostate cancer progression and decrease the therapeutic effect of docetaxel or enzalutamide [8]. B7-H3 (also known as CD276), a B7 family immune checkpoint, can be a promising target for PCa immunotherapy, particularly in the early weeks post-ADT before PCa enters dormancy [9]. Recently, active Stat5 signaling, a known promoter of prostate cancer growth and clinical progression, was unexpectedly found to be a potent inducer of AR gene transcription in PCa, which indicates that pharmacological Stat5 inhibitors may represent a new strategy for suppressing ARs and CRPC growth [10].

With the development of DNA microarrays and high-throughput sequencing, it is both efficient and effective to explore key gene modules related to tumor progression via the use of bioinformatics techniques and big data integration. The weighted correlation network analysis (WGCNA) is an efficient and accurate bioinformatics method for analyzing microarray data that can be used to systematically investigate highly synergistically altered gene modules [11]. The WGCNA divides genes into several modules based on the similarity of gene expression profiles and can be used to identify the relationship between gene sets and clinical characteristics. WGCNA methods have been successfully applied to identify key gene modules in many cancers, including bladder cancer [12], ovarian cancer [13], and breast cancer [14]. However, the primary objective of WGCNA aims to tell the difference between the “disease state” and “normal state” and may fail to accurately predict the early onset of disease before its development. To overcome this bottleneck, the dynamic network biomarker (DNB) theory, based on the dynamic features of molecules within the biological system, was proposed [15]. The DNB theory is a kind of nonlinear dynamics theory which aims to find a group with high correlation and strong collective fluctuations that affect the dramatic changes in diseases. DNBs reveal early warning signals of critical transitions before the deterioration of complex diseases. The DNB method has been applied to real disease datasets and has been used to identify the pre-disease states of several cancers, such as hepatocellular carcinoma [16], colon and rectal cancer [17], and liver cancer [18]. These studies provide us with a reliable and robust technical basis for identifying critical signals, as well as prognostic indicators, in CRPC.

In this study, we combined the methods of DNBs and the WGCNA to explore novel biomarkers in androgen deprivation therapy resistance and the prognosis of prostate cancer. The flowchart of this study is shown in Figure 1. By analyzing the dynamic changes in key modules involved in androgen-associated activities in CRPC and identifying stage-related gene modules combined in the weighted gene co-expression network, we identified four genes, including *SCD*, *NARS2*, *ALDH1A1*, and *NFXL1*, as core members of DNBs, which can serve as biomarkers involved in the transition of prostate cancer cells from an androgen-dependent state to a castration-resistant state. This study is expected to provide novel diagnostic and therapeutic targets for prostate cancer patients treated with androgen deprivation therapy and offer new insights into the molecular pathology of CRPC progression from dynamic network perspectives.

## 2. Materials and Methods

### 2.1. Data Acquisition and Preprocessing

The RNA sequencing data and corresponding clinical data of prostate cancer samples were acquired from The Cancer Genome Atlas (TCGA) (http://cancergenome.nih.gov/ accessed on 15 February 2024). The single-cell RNA sequencing (scRNA-seq) information of GSE137829 was obtained via the Gene Expression Omnibus (GEO) database (https://www.ncbi.nlm.nih.gov/ accessed on 9 December 2023). CRPC bulk RNA-seq data were obtained from 3 GEO datasets (GSE700770, GSE80609, and GSE111177). TCGA-PRAD data were downloaded from the UCSC Xena database (https://xenabrowser.net/datapages/ accessed on 12 March 2024).

Based on the description in the article, a total of 14 CRPC patients were selected from the CRPC cohort according to their clinical information, and the corresponding single-cell RNA matrix data of 13 CRPC patients were found and downloaded from the GSE70770 dataset. In GSE80609, 12 CRPC patients were identified from the NGS cohort based on their clinical characteristics and the corresponding raw matrix data were downloaded. In GSE111177, we screened and selected 20 CRPC patients who underwent ADT based on their clinical information provided by the original article.

On the basis of the 2021 Canadian Urological Association (CUA)–Canadian Uro-Oncology Group (CUOG) guideline, for the management of castration-resistant prostate cancer (CRPC), we defined the “disease progression” of CRPC as “Deterioration occurred in the patient after receiving treatment, including disease worsening despite castrate levels of testosterone, the occurrence of new lesions, the progression of pre-existing disease, and/or the appearance of new metastases” in this study.

We applied the R package “Seurat (V4.0)” to process scRNA-seq data and conduct cell type annotation. We excluded cells with fewer than 200 or more than 6000 detected expressed genes (where each gene had to have at least one unique molecular identifier aligned in at least three cells). Cells with more than 10% expression of mitochondrial genes were excluded to remove low-activity cells.

We performed logarithmic normalization separately on the data from 6 samples. The “FindVariableFeatures” function was used to identify highly variable genes (based on variance stabilizing transformation, “vst”), followed by the removal of batch effects from the samples using the “FindIntegrationAnchors” function of the canonical correlation analysis (CCA) method. Additionally, we integrated the data using the IntegrateData function and scaled all genes using the ScaleData function. Principal component analysis (PCA) was employed to conduct dimensionality reduction and identify anchors. We selected dim = 30 and clustered cells using the “FindNeighbors” and “FindClusters” functions (resolution = 0.5), resulting in 21 clusters. Furthermore, we downloaded marker genes and related data for human cells from CellMarker (http://bio-bigdata.hrbmu.edu.cn/CellMarker1.0/, 17 September 2024) and manually annotated the cells [19].

### 2.2. Bulk RNA-Seq Data Processing

We downloaded the series matrix files and their platform annotation information and eliminated more than half of the sample values or probes that detected multiple genes for analysis. Due to the differences in gene symbols across different microarray platforms, the probes were assigned to their Entrez identifiers according to each platform’s annotation file. The R package “limma” was employed to identify differentially expressed genes (DEGs). We used the arithmetic mean and integrated groups to interpret the gene expression level if multiple probe groups corresponded to the same Entrez ID. Then, the differentially expressed genes (DEGs) were normalized by the number of reads per sample, and the ensemble IDs were converted to gene IDs by the R package “biomartr” [20]. We selected 13 CRPC patient samples from the GSE70770 dataset, 12 from the GSE80609 dataset, and 20 from the GSE111177 dataset from patients who received androgen deprivation therapy (ADT). For the TCGA dataset, genes with an expression level of 0 (not detected) in more than 10% of patients were excluded from further analysis. We combined clinical information from the TCGA database to identify patient samples that underwent ADT.

### 2.3. Pseudotime Trajectory Analysis

The “Monocle2” R package (version 2.20.0) was employed to elucidate the epithelial cell developmental trajectory and characterize the functional change processes and identify potential lineage differentiation between clusters. Based on the machine learning method of “reversed graph embedding”, it can automatically infer the trajectory from high-dimensional RNA-seq data. In this study, we constructed the developmental trajectory based on the following steps. First, the core genes in each cluster were identified by the “Differential Gene Test” function. Then, the expression spectrum was simplified by the “reduceDimension” function and the “DDRTree” method (max_components = 2) to construct a minimum spanning tree (MST), which represents the potential path of cell differentiation. Then, the “orderCells” function was used to sort cells and assign pseudotime values. After this, branch expression analysis modeling (BEAM) was employed to detect and analyze specific branch points and identify genes with branch-dependent expression based on the pseudotime value.

### 2.4. Dynamic Network Biomarker (DNB)

The transition process of disease can be roughly divided into three states, the before-transition state, critical state, and after-transition state. The genes that only appear in the critical state and play key roles in the critical state can be identified as DNBs. In this work, we applied the R package “BioTIP” to identify dynamic network biomolecules and predict the differentiation trajectory of the prostate epithelium [21]. Dynamic network biomarkers (DNBs) are used for biological tipping-point characterization and focus on the detection and assessment of different stages of disease. It is a time-dependent method [22] which studies the location changes in the markers over time and the relationships among network markers over time. The DNB method follows three major criteria: (1) the standard deviation of the DNB molecule group increases dramatically in the critical state; (2) the correlation between any two molecules in the DNB molecule group increases significantly in the critical state; (3) the correlation between the DNB molecule group and the other group decreases steeply in the critical state.

In this study, we performed a DNB analysis according to the following steps: (1) data preparation and preprocessing: we extracted the expression data of different development trajectory states and removed genes with mean cell expression values less than 0.01; (2) estimating the random IC scores by permuting the expression values of genes: we applied the “Simulation_Ic” function to calculate a random index of critical transition and randomly filtered 300 genes and ran it 1000 times to calculate IC scores for each state of every cell subset; (3) filtering a multi-state dataset based on a cutoff value for standard deviation per state and optimization: we used the “optimize.sd_selection” function to select the top 1% transcripts and randomly selected 80% of samples and calculated it 100 times to select the filtered expression dataset matrix with the highest standard deviation; and (4) building node networks: We applied the “getNetwork” function to construct a correlation network for each trajectory state. Pearson’s coefficient analysis was used to identify significantly correlated genes (*p* < 0.1) in the co-expression network. Then, we used the “getCluster_methods” function to extract genes from each subnetwork (module). (5) We identified critical transition signals (CTSs) used in the DNB module. We calculated the module key index (MCI) for each trajectory state in the dataset using the “getMCI” function and used the “getMaxMCImember” function to filter out the top 3 modules in each trajectory state. (6) Finding the tipping point and evaluating the CTS: We first recorded the maximum MCI of candidate modules at different trajectory states, and then extracted the top 2 modules according to the MCI scores. After this, we estimated the correlation matrix using the “cor.shrink” function, followed by a recalculation of the critical transition random index (IC) using the “simulation_Ic” function for each module gene; (7) verifying using the IC score: We estimated the random IC scores by permutating the expression values of genes and returned to the observed IC (red) and simulated IC scores (gray) for a given state. Then, we estimated the random IC scores by randomly shuffling the cell labels. We evaluated the random score from the shuffling sample labels and excluded natural sample correlations within phenotypic states (cell subsets) and returned the score to the IC of observed (red) and simulated IC scores (gray) for a given state.

### 2.5. Gene Set Variation Analysis (GSVA)

Gene set variation analysis (GSVA) is a nonparametric and unsupervised method for estimating the enrichment of gene sets in transcriptomic data [23]. By performing comprehensive scoring on the gene sets of interest, GSVA converts changes at the gene level to those at the pathway level, subsequently determining the biological functions of the sample. In this study, we employed the GSVA algorithm to score the identified DNB module gene sets and to evaluate the potential biological function changes across different modules.

### 2.6. CellChat Analysis

We employed the “CellChat” R package (v1.6.1) for the cell interaction analysis [24]. We analyzed the possible interactions among epithelial subgroups based on the data of ligand–receptor pair data in the CellChatDB.

### 2.7. Weighted Correlation Network Analysis (WGCNA)

We applied the “WGCNA” R package to perform WGCNA according to the following steps [25]. (1) Define the similarity matrix. (2) Select the weight coefficient β = 12 and convert the similarity matrix into an adjacency matrix. (3) Convert the adjacency matrix into a topological overlap matrix (TOM). (4) Perform hierarchical clustering of data based on the TOM to obtain a hierarchical clustering tree. (5) Use the dynamic tree-cutting method to identify modules from the hierarchical clustering tree. (6) Calculate the module eigengenes (MEs) for each module, where MEs represent the overall expression level of the module. We calculated the Pearson correlation coefficient between the MEs of each module, and the Pearson correlation coefficient was defined as the average distance between MEs of each module. We applied the average linkage hierarchical clustering method to cluster all MEs of the modules, with a minimum value (genome) set to 100 and combined modules with high similarity to obtain a co-expression network.

### 2.8. Gene Enrichment Analysis

We downloaded a dataset related to the cancer hallmark and gene oncology (GO) pathway from the MisgDB website (https://www.gsea-msigdb.org/gsea/msigdb, access on 23 February 2024) [26]. The “clusterProfiler” R package [27] was used to perform the gene pathway enrichment analysis.

### 2.9. PPI and Fuzzy Clustering

To assess the strength of the correlations between genes in the DNB module, we downloaded protein–protein interaction pairs from Homo sapiens from the STRING database and filtered them for those with combined scores greater than 400, constructing a DNB module-based interaction network. We then selected all the interacting genes from the network. Considering the difficulty in separating gene expression trends, we used a noise-robust soft clustering method [28]. We applied a fuzzy c-means clustering method (FCM) based on the time trend to classify genes with similar expression patterns into clusters. The R package “Mfuzz” was employed to conduct analysis. The clustering parameter was set to 6.

### 2.10. Construction of the Risk Prediction Model and Nomogram

We used the following formula to calculate the risk score for each patient. “RiskScore = gene Exp1 × β1 + gene Exp2 × β2 + gene Exp3 × β3 + … + gene Expi × βi”.

In the formula, “gene Expi” refers to the gene expression level, while β refers to the correlation coefficient of ligand–receptor pairs in the multivariable Cox regression analysis. We applied the “surv_cutpoint” function to evaluate thresholds and divided patients into “high-risk” and “low-risk” groups and used the Kaplan–Meier method to plot survival curves for prognostic analysis. The log-rank test was performed to determine the significance of the differences. Patient survival curves and risk maps were visualized by the R packages “survminer” and “ggrisk” [29,30]. The ROC curves were plotted using the “survROC” R package.

To further evaluate the robustness of the risk prediction model, we combined patient information with detailed clinical and pathological outcomes including age, Gleason score, and pathological tumor stage from the TCGA dataset to construct a nomogram evaluation model [31]. The serum PSA level was excluded, as there were few patients whose preoperative PSA level was high (PSA > 10 ng/mL). All clinical and pathological characteristics were considered as categorical variables for evaluation via the nomogram analysis.

### 2.11. Immunohistochemistry (IHC)

IHC staining was performed on the samples from 20 hormone-sensitive PC patients and 10 CRPC patients. The study was approved by the Institutional Review Board of the School of Life Sciences, Central South University (approval number: IRB 2024-1-43), and compliant with recommendations from the Declaration of Helsinki for biomedical research involving human subjects. The samples were fixed in 4% neutral buffered paraformaldehyde, embedded in paraffin, and cut into 5 μm slices. After deparaffinization, hydration, and antigen retrieval, these sections were incubated with a corresponding primary antibody, followed by incubation with a biotinylated secondary antibody. Sample tissues were probed with antibodies against SCD, ALDH1A1, NARS2, and NFXL1 at a 1:100 dilution following standard IHC protocol. The primary antibodies were anti-SCD (, dilution 1:100, Abcam, Cambridge, MA, USA), anti-ALDH1A1 (dilution 1:100, Proteintech, Wuhan China), anti-NARS2 (dilution 1:100, Abcam, Cambridge, MA, USA), and anti-NFXL1 (dilution 1:100, Abcam, Cambridge, MA, USA). For cell visualization and imaging, the KF-PRO-400-HI high-throughput digital pathology slide scanner (KFBio Inc., Ningbo, Zhejiang, China) was used. For the immunoreactive score of each gene, a staining index (values, 0–12) was determined by multiplying the score for staining intensity with the score for positive area. The intensity of staining was scored as follows: 0, negative; 1, weak; 2, moderate; and 3, strong. The frequency of positive cells was defined as follows: 0, less than 5%; 1, 5% to 25%; 2, 26% to 50%; 3, 51% to 75%; and 4, greater than 75%. When the staining was heterogeneous, each component was scored independently and summed for the results. For example, a specimen containing 80% tumor cells with moderate intensity (4 × 2 = 8) and another 20% tumor cells with weak intensity (1 × 1 = 1) received a final score of 8 + 1 = 9. For statistical analysis, scores of 0 to 7 were considered low expression and scores of 8 to 12 were considered high expression. The protein expression was scored by two independent pathologists who lacked prior knowledge of the patients’ clinicopathological characteristics (double-blinded). In cases of discrepant results, the values were discussed until an agreement was reached.

### 2.12. Statistical Analysis

All data analyses were conducted on the R platform (version 4.3.0). Student’s *t*-test or the Wilcoxon rank-sum test were used to compare continuous variables between two subgroups. One-way ANOVA or the Kruskal–Wallis test was used to compare differences among the three groups. Pearson correlation was used to assess the correlation between normally distributed variables, while Spearman correlation was used to analyze nonnormally distributed variables. The Benjamini and Hochberg (BH) method was applied to estimate the false discovery rate of multiple tests. The “survminer” R package [32] was used to perform Kaplan–Meier analysis and log-rank tests to evaluate survival differences among groups.

## 3. Results

### 3.1. Segmentation and Trajectory Inference of CRPC Epithelial Cells

Epithelial cells play vital roles in cancer tissues, and changes in epithelial cells in tumor tissues often suggest the occurrence and progression of cancer. To study the features of the epithelial cell group of patients with castration-resistant prostate cancer (CRPC), scRNA-seq profiles of 23,987 cells were collected. The samples of six CRPC patients from the GSE137829 dataset were merged. The R package “Seurat(V4.0)” was employed to process scRNA-seq data and annotate the cell type. Furthermore, we applied the “FindCluster” function to select an optimal cell resolution of 0.5 for clustering, and obtained a total of 21 clusters (Figure 2A,B). Then, 21 clusters were manually annotated into 10 types of cells by different cell markers (CD4 T cells, CD8T cells, B cells, plasma cells, myeloid cells, mast cells, epithelial cells, endothelial cells, fibroblasts, and pericytes) (Figure 2D). The expression of cell marker genes is shown in Figure 2C. We found that the percentage of each type of cell subtype among the sample of each patient was roughly the same in the dataset, and the most prominent subtype is epithelial cells. Endothelial cells and fibroblasts accounted for a greater proportion of cells in some patients (Figure 2E).

To analyze the cytodifferentiation characteristics of epithelial cells in patients with CRPC, the “subset” function was applied to select 11,143 epithelial cells and reclusters. The “Findcluster” function with resolution set to 1.2 was used for clustering, resulting in 15 clusters (Appendix A). The epithelial cells were reannotated into four cell types by epithelial cell subgroup markers (Figure 3A) (basal cells, luminal cells, neuroendocrine cells, and other cells), in which clusters 0, 1, 2, 3, 5, 7, 9, 10, 11, 12, 13, and 14 were labeled as luminal cells, while cluster 4 was labeled as other cells. Cluster 6 was labeled as neuroendocrine cells and cluster 8 was labeled as basal cells. The cell marker gene expression levels are shown in Figure 3C. The proportions of epithelial cell subgroups were similar across different samples, with the most significant proportion being the luminal subgroup (Figure 3B). Luminal cells have been identified as the primary cell type associated with prostate carcinogenesis and development [33].

To better understand the molecular mechanism underlying the occurrence of castration resistance, we applied the R package “Monocle2” to construct a developmental trajectory of CRPC epithelial cells. The trajectory demonstrates a continuous transition process of cellular differentiation states in epithelial cells, revealing the dynamic process of epithelial cell development. The results showed the state of cellular pseudotime development, differentiation states, and the sample distribution among different epithelial subtypes (Figure 3D,E and Appendix A). The trajectory was divided into four parts (S1, S2, S3, and S4) based on the trajectory nodes. According to a previous study [33], three important gene markers (TACSTD2, KRT4, and PSCA) originating from luminal cells were expressed in deteriorating trajectories (Appendix A) and can be recognized as the beginning of pseudotime development. The level of androgen receptors (ARs), an important marker of castration-resistant prostate cancer, gradually increases with the continuous differentiation of epithelial cells (Appendix A), suggesting that the differentiated cells of tumor epithelial cells gradually develop into castration-resistant cells [34].

### 3.2. Prediction of Key Transition Subgroups and Gene Modules in Epithelial Cells

The function of epithelial cells at different stages may change with the cellular differentiation of epithelial cells. We applied dynamic network biomarker (DNB) methods to clarify the status of epithelial cells in prostate cancer patients who develop castration resistance after androgen deprivation therapy (ADT). The “BioTIP” package was used to perform the DNB analysis. First, we identified critical tipping points of cellular differentiation states in different epithelial cells, calculated the corresponding critical transition random index (IC scores), and filtered the expression spectral matrix. Through the correlation analysis, we constructed a gene co-expression network module across various differentiated cellular states and identified the hypothesized critical transition signals (CTSs) in CRPC epithelial cells (Figure 4A). Next, we estimated the gene correlation matrix for each module and recalculated the critical transition random index by randomly perturbing the gene tags (Figure 4B,C) and sample tags (Figure 4D,E). Finally, the significance of different modules was calculated. The most significant transition signal conversion DNB module was identified in the cellular differentiation state S2. The DNB module consists of 32 genes, and through gene expression analysis, it was discovered that all genes in the module have a greater expression level than genes in other epithelial cell subgroups (Figure 4F).

To clarify the functional differences in the DNB module across different epithelial cell differentiation states, we applied the GSVA (gene set variation analysis) method to calculate the enrichment matrix for the GO terms and the cancer hallmark pathways. The results showed that the DNB module gene sets were mainly enriched in androgen-related GO pathways, including the androgen metabolic signaling pathway, the androgen receptor signaling pathway, and the androgen biosynthetic process (Figure 4G). Similarly, the DNB module showed significantly greater gene expression in the androgen response, protein secretion, and mTORC1 signaling pathways of the cancer hallmark (Figure 4H,I), which revealed that the DNB module may play a crucial role in transition signals during the development and occurrence of CRPC in patients. Therefore, it is likely that this epithelial cell subgroup possesses a dynamic gene expression module with key transitional state functions and unique expression patterns.

### 3.3. Cellular Communication in Key Subgroups of Epithelial Cells

We explored differences in the functions of epithelial cells in critical transition states based on CRPC scRNA-seq data. First, based on the results of epithelial cell differentiation, the “CellChat” R package was applied to conduct a cellular communication analysis in epithelial cells in both the S2 and S1 stages. The total number and strength of epithelial cells in S2 were slightly greater than those in S1 (Figure 5A and Appendix A). Further analysis was conducted to clarify the communication patterns between epithelial cells and other cell subsets. The results revealed that the number and strength of pericyte cells and epithelial and fibroblast cells that communicated with epithelial cells were increased in the S2 subgroup (Figure 5B and Appendix A).

Subsequently, we examined the cellular communication pathways of S2 and S1 epithelial cells. After performing dimensionality reduction based on functional similarity among different signaling pathways, we discovered that pathways such as EPHA1, SEMA6, and others exhibited the most significant functional differences between the two groups (Appendix A). SEMA6 can promote angiogenesis [35]. The EPHA pathway contributes to the stimulation of ARs through inducing expression of proto-oncogenes [36], and could increase the invasion of CRPC [37,38].

Moreover, we compared the differences in cell communication between the S2 epithelial cell subgroups and other cell types and found that the increased signal intensity occurred mainly in signaling pathways such as BMP, DESMOSOME, GDF, and PGDF (Figure 5C). To further explore differences in these signaling pathways, we investigated pathway alterations between the two groups. The BMP signaling pathway mainly involved the transduction of signals from fibroblasts in the S2 subgroup to epithelial cells, whereas the DESMOSOME signaling pathway mainly changed in the internal signaling within the epithelial cells in the S2 subgroup. Changes in the GDF signaling pathway were mainly related to intercellular signaling between S2 epithelial cells, fibroblasts, and endothelial cells. Changes in the PGDF signaling pathway primarily occurred in the signal transmission from epithelial cells to fibroblasts and pericytes within the S2 subgroup (Figure 5D and Appendix A). The BMP signaling pathway can be involved in various developmental processes, including cell proliferation, cellular differentiation, apoptosis, and angiogenesis [39], and the DESMOSOME signaling pathway can stimulate and enhance cell adhesion [40]. Finally, we observed specific differential ligand–receptor interactions between the two groups, where 75-fold greater numbers of ligand–receptor interactions were found in ligands between epithelial cells from group 1 (S1) and group 2 (S2), which act on other cells. The expression of MDK- and APP-related ligand–receptors in the S2 subgroups significantly increased in epithelial cells and most of the other cells, whereas the expression of MIF and other ligand–receptors significantly decreased in epithelial cells and most of the other cells (Figure 6A,B). The MDK signaling pathway plays a role in driving castration resistance and has been previously identified in CTCs [41]. The APP signaling pathway participates in the regulation of androgen and is related to the binding of the AR gene [42]. However, the MIF signaling pathway can inhibit prostate cell growth, invasion, and the inflammatory response [43]. Cell communication of COL1A1 and other ligand-receptors discovered in other cell subgroups significantly increasd in ligand–receptors in the lower score group, while MDK ligand–receptors significantly decreased (Figure 6C,D).

To further explore the significance of S2 epithelial cells in cellular communication, we applied “CellChat” analysis to the subgroups of epithelial cells from S1 to S4. We found that the subset of S2 epithelial cells had relatively stronger interactions with pericytes and fibroblasts in the PGDF signaling pathway than with the other subsets of epithelial cells (Appendix A), whereas the FGF signaling pathway was mediated by fibroblasts on S2 epithelial cells (Appendix A). The PGDF signaling pathway is primarily composed of platelet-derived growth factor-related genes and is a crucial regulatory factor for mesenchymal cells. These genes are often expressed in relation to aggressiveness, tumor size, chemotherapy resistance, and the clinical recurrence of prostate cancer [44,45]. The fibroblast growth factor signaling pathway is mainly composed of genes related to fibroblast growth factors, which can influence the progression of prostate cancer through the interactions between epithelial and stromal components [46]. By identifying the unique communication within the subgroup of S2 epithelial cells, our results suggested changes in related signaling pathways and their potential impact on tumor progression and recurrence.

### 3.4. Construction of the Co-Expressed Gene Module Associated with Androgen Regulation via Bulk RNA-Seq of CRPC Cells

The interactions between genes resemble cellular interactions. To explore the potential associations between gene expression patterns in CRPC patients, we applied the WGCNA method to further analyze the gene co-expression profiles in CRPC patients. First, we obtained two gene expression datasets of prostate cancer patients, GSE70770 and GSE80609, from the GEO database. After performing data preprocessing based on clinical information labels, the gene expression profiles of CRPC patients were extracted from two datasets. Next, we performed the weighted gene co-expression network analysis (WGCNA) on the two datasets. In this study, the co-expression network was classified as an unstructured network, and we calculated the optimal soft threshold for near neighbors. The expression matrix was converted into a neighbor matrix, and then the neighbor matrix was transformed into a topological matrix. Based on this topological matrix, the averaging linkage clustering method was used to cluster genes. Following the standard of the hybrid dynamic shear tree, the number of genes in the module was set to 100 as the minimum number in each module. We performed a clustering analysis on the modules and merged modules with close distances into new modules. Finally, 15 module clusters were identified in GSE70770 (Figure 7A and Appendix A) and 13 module clusters were identified in GSE80609 (Figure 7B and Appendix A). Each module consisted of genes with similar expression patterns.

After performing a cancer hallmark pathway enrichment analysis on the two datasets, we discovered that the “midnightblue” module in the GSE70770 dataset and the “blue” module in the GSE80609 dataset were significantly enriched in the androgen response hallmark signaling pathway and protein secretion hallmark signaling pathway (Figure 7C,D). It will stimulate the secretion of androgen and proteins in cancer tissues in the corresponding cancerous tissue at the same time. This may increase the probability of immune evasion of the epithelial cells, which suggests that these two modules may be potentially associated with tumorigenesis and tumor development in CRPC [47].

### 3.5. Identification of Key Biomarkers for Castration Resistance in PCa

To explore the interactions among DNB module genes and their effects on various biological processes, we combined protein interaction information in STRING and constructed a PPI network of the DNB module, where 30 genes in the DNB module have interacted with other genes. Then, we extracted genes from the protein interaction network and applied the soft clustering algorithm to classify the gene sets according to the expression trend in the DNB module and its neighboring gene classification (Figure 8A). The results showed that cluster 2 and cluster 4 gene sets had greater expression changes in the S2 epithelial cell subgroup. Subsequently, we combined the enrichment analysis outcomes of the two cluster gene sets; the cluster 2 gene set was more likely to be enriched in cancer-related pathways such as DNA repairment and oxidative phosphorylation (Appendix A). The cluster 4 gene set was more likely to be enriched in cancer-related pathways such as androgen response pathways (Figure 8B). Moreover, the GO enrichment analysis showed that the cluster 4 gene set was significantly correlated with numerous metabolic and biosynthetic processes (Appendix A). The results revealed that the cluster 4 gene set plays an important role in the process of castration resistance in prostate cancer (Figure 8C).

Combining our findings from the WGCNA of bulk RNA-seq datasets, we identified a set of gene modules associated with androgen-responsive pathways. In parallel, we applied the DNB method to clarify the condition of epithelial cell subgroups in scRNA-seq datasets and found the cellular differentiation state S2 as the most significant transition signal conversion DNB module, which consists of 32 genes. We then explored the co-expression modules of the key transformations in the DNB module and the WGCNA module using the soft clustering analysis (R package “Mfuzz”), and finally identified four genes (*SCD*, *NARS2*, *ALDH1A1*, and *NFXL1*) that may be related to castration resistance development in PCa patients. On the one hand, studies have shown that the expression of *SCD* (stearoyl-CoA desaturase) has significantly increased, indicating *SCD* can be considered as a potential treatment target [48,49]. On the other hand, SCD can promote the proliferation of androgen receptor-positive LNCaP cells, enhance the transcriptional activity of ARs induced by dihydrotestosterone (DHT), and result in the increased expression of prostate-specific antigens (PSAs) and kallikrein-related peptidase 2 (*KLK2*) [50]. Aldehyde dehydrogenase 1 family member A1 (*ALDH1A1*) is not only a marker for malignant prostate stem cells but can also serve as a predictor of prognosis in PCa patients [51]. Its high expression is associated with the development of prostate cancer [52], suggesting that this gene may play a role in the progression of prostate cancer.

### 3.6. Construction and Evaluation of the Prognostic Risk Model

To verify the progression-free interval (PFI) evaluation of gene status in the context of androgen deprivation therapy (ADT) for PCa patients, we performed a multivariable Cox regression analysis to fit the four genes (*SCD*, *NARS2*, *ALDH1A1*, and *NFXL1*) into an initial model. For each patient, the risk score (Figure 9A) was calculated by multiplying the gene expression level by the corresponding regression coefficients derived from the multivariable Cox regression model. The risk score in this case is = (0.3111 × *ALDH1A1* expression level) + (0.0087 × *SCD* expression level) and (−0.10137 × *NARS2* expression level) + (−0.4833 × *NFXL1* expression level). In the TCGA training cohort, a best-fitting threshold was applied to group patients into low- and high-risk groups. The Kaplan–Meier analysis results showed that patients in the low-risk group had significantly longer progression-free intervals (PFIs) than those in the high-risk group (Figure 9B). In the GSE111177 validation cohort, we used the same risk score model to grade 20 patients on progression risk with the best cutoff value. By grouping patients into low- and high-risk groups, the Kaplan–Meier analysis indicated that the high-risk group had shorter recurrence times (Figure 9C).

To develop a quantitative method for predicting the progression-free interval (PFI), we combined the progression risk score with other clinicopathological characteristics, including the age at diagnosis, Gleason score, and tumor stage at biopsy, in the TCGA training cohort. For each factor, we calculated a point and then obtained the total points for all factors to create the nomogram of the TCGA training cohort to evaluate the overall PFI rate. Additionally, we created receiver operating characteristic (ROC) curves and calibration curves to evaluate the reliability of the nomogram, which showed the relationship between clinical sensitivity and specificity across different cutoff points. We performed a Cox multivariate regression analysis on the clinical data and the risk score. The results showed that the risk score was an independent prognostic factor for the PFI in the TCGA dataset (hazard ratio (HR) = 2.71, 95% CI = 1.39 to 5.28, * *p* < 0.05). Based on the results of univariate and multivariate analyses, we constructed a nomogram model combining clinical characteristics and the risk score (Figure 9D). In the model, the risk score-based features had the greatest impact on survival prediction. As depicted in Figure 9E and Appendix A, the 1-, 2-, and 3-year survival nomogram calibration curves were in good agreement with the standard curve. Furthermore, the areas under the ROC curves (AUCs) for the 1-year, 2-year, and 3-year PFI predictions were 0.694, 0.748, and 0.717, respectively (Figure 9F). In summary, this risk model can reasonably predict the prognosis of prostate cancer patients treated with androgen deprivation therapy (ADT).

### 3.7. IHC Analysis

The IHC analysis showed that compared to HSPC samples, ALDH1A1 and SCD protein expression levels were significantly higher in CRPC patients, but the protein expression level of NARS2 was relatively lower Figure 10A,B. These data further validated the risk score of the genes in the nomogram and suggested that both ALDH1A1 and SCD can be risk factors, while NARS2 may be a protective factor in CRPC progression.

## 4. Discussion

This study comprehensively analyzed key gene modules in CRPC cells through the integration of single-cell and bulk RNA sequencing data combined with the methods of DNBs, a WGCNA, and noise-robust soft clustering. By combining our findings from the DNB method in the scRNA-seq dataset and the WGCNA in bulk RNA-seq datasets, we identified a set of key biomarkers associated with androgen-related pathways, including four key genes (SCD, NARS2, ALDH1A1, and NFXL1). Finally, we developed a risk prediction model combining the risk scores of the four key genes and other clinicopathological characteristics to assess the prognosis of patients treated with androgen deprivation therapy (ADT). The TCGA training cohort demonstrated that this risk score model reliably evaluated the PFI of prostate cancer patients treated with androgen deprivation therapy (ADT). The external GEO cohort (GSE111177) also validated the high performance of this risk prediction model. An IHC analysis was conducted to compare the protein expression level of the genes in CRPC and HSPC patient tissues. Overall, our findings show that SCD, NARS2, ALDH1A1, and NFXL1 are key biomarkers associated with androgen-related signaling pathways in CRPC.

Nomograms have been widely used by oncologists to generate prognostic information for individual patients due to their numerical probability and user-friendly interface [53,54,55]. In this study, a novel nomogram was established by integrating the risk score of four genes (SCD, ALDH1A1, NARS2, and NFXL1), age, Gleason score, and tumor stage, each of which was an independent prognostic factor according to the multivariate Cox regression analysis. This risk score model showed the ability to predict the prognosis of prostate cancer patients treated with androgen deprivation therapy (ADT). This risk score model is superior to traditional clinical factors for prognosis evaluation. First, it can quantify the risk score of PCa patients treated with ADT; a higher risk score represents a higher chance of tumor progression. Second, the predictive ability of the risk score was also better than that of other clinical variables, as exemplified by the highest AUC of 0.748 in the ROC curves. Third, the genes in the risk score model were identified by DNBs and the WGCNA and further filtered by univariate Cox and multivariate Cox regression analyses. This prognostic model relied on fewer genes but retained good performance for predicting patient prognosis. Fourth, the reliability of the risk score model was validated in the GSE111177 validation cohort. Overall, the proposed risk score model may be useful for the prognostic evaluation of prostate cancer patients treated with ADT.

With respect to the genes in the risk score model, we found that SCD and ALDH1A1 may be potential risk genes, while NARS2 and NFXL1 can be favorable prognostic genes. Stearoyl-CoA desaturase (SCD) is an enzyme that controls the synthesis of unsaturated fatty acids and is essential in breast and prostate cancer cells. SCD has been shown to promote proliferation and disease progression in prostate cancer by affecting cellular signaling cascades and modulating androgen receptor transactivation [56]. A functional genomics analysis showed that SCD inhibition altered the cellular lipid composition and impeded cell viability in the absence of exogenous lipids in prostate cancer cells. SCD inhibition also altered the cardiolipin composition, leading to the release of cytochrome C and the induction of apoptosis [57,58]. This is in line with our findings from the CellChat analysis that signaling pathways which are associated with apoptosis were more active in the subpopulation of S2 epithelial cells than in other cells. The aldehyde dehydrogenase 1A1 (ALDH1A1) isoform, which can positively regulate tumor cell survival in circulation, extravasation, and metastatic dissemination, is correlated with Aldefluor activity in PCa patients’ tissue specimens [52]. PCa cells with high ALDH activity were previously characterized as a population with high metastasis-initiating properties [59]. ALDH1A isoform members have generated considerable interest, as ALDH1A1 has frequently been shown to be expressed in prostate cancer stem cell populations and may contribute to malignancy [60]. Recently, in vivo models confirmed that ALDH1A1 plays as a positive regulator of metastatic dissemination in the regulation of PCa metastases [52]. Higher ALDH1A1 and SCD expression was found in CRPC patients compared with HSPC groups in our IHC analysis, which was consistent with previous studies implying that higher SCD and ALDH1A1 expression levels were closely associated with CRPC progression [60,61].

NARS2 (asparaginyl-tRNA synthetase 2, also known as asnRS) is a nuclear gene encoding AsnRS that functions in mitochondria [62]. Biallelic mutations in NARS2 have been recently identified in patients with hearing impairment, intellectual disability, seizures, hypotonia, delayed neurodevelopment, renal dysfunction, and/or liver involvement [63]. NARS2 variants can disrupt the integrity of the mitochondrial protein synthesis, which is essential and fundamental for the mitochondrial oxidative phosphorylation complex, thus influencing cellular energy production [64]. Recently, research has shown that NARS2 has an important impact on immune resistance and drug resistance in melanoma [65]. Yet, few studies have demonstrated the role of NARS2 in prostate cancer. The IHC analysis in our study suggested that the protein expression level of NARS2 might decrease with the progression of CRPC. This may be one of the few studies to provide some evidence about the role of NARS2 in prostate cancer progression and yet, more in vivo analysis is needed to clarify the role of such a gene. The NFXL1 gene encodes an NFX-1-type nuclear zinc finger transcriptional repressor that is expressed in the cytoplasm [66]. NFXL1 is socalled because it is a paralog of the NF-X1 transcription factor which binds the X-box sequence of class II MHC genes [67]. This feature may be relevant according to a study that revealed an association between human leukocyte antigen (HLA) loci and specific language impairments [68]. However, until now, little is known about the function of the NFXL1 protein, nor have disorders been identified that arise from the mutation of this gene; additionally, no animal knockouts have been described.

At present, the prognostic value of these genes in the risk core model has been evaluated in PCa patients treated with ADT, which will hopefully provide novel biomarkers for future studies on molecular insights into PCa. Based on the risk score and clinical factors, including age, Gleason grade, and tumor stage, we constructed a nomogram for precisely evaluating the patients’ 1-, 2-, and 3-year PFIs. A higher score calculated from this nomogram represents a greater chance of deterioration. Integrating the risk score with clinicopathological factors will improve the accuracy of PFI prediction. This may provide crucial information for the individual management of PCa patients treated with ADT. In general, this risk score model and nomogram will be helpful for evaluating the prognosis of PCa patients treated with ADT.

While general biomarkers usually treat patients on the basis that the disease has already occurred, DNBs are a group of biomolecules with strong dynamic correlation, and their molecular concentrations undergo dynamic changes rather than maintaining a constant value for the critical state [15]. DNBs reveal early warning signals of critical transitions before the disease deteriorates. Taking advantage of this technique, pathophysiological changes at different stages and periods can be dynamically displayed in a time series through DNBs, and stage-specific and severity-specific biomarkers of prostate cancer patients can be identified [69]. Thus, patients can be treated by adjusting the role of DNBs in disease through gene targeting therapy and other methods before the cancer progresses further. The key modules in DNBs may play an important role in the diagnosis and treatment of the disease. However, the specific application of DNBs still needs to be repeatedly validated by a series of both in vitro and in vivo experiments, and the development of such scientific tools and clinical practices still needs time to be explored and clarified. Our study found that DNBs and four genes (SCD, NARS2, ALDH1A1, and NFXL1), as relevant factors associated with androgen-related signaling pathways in prostate cancer cells, can be further used as indicators of PCa progression after androgen deprivation therapy. Although the prediction of related gene expression is not currently feasible, we anticipate that with the advancement of artificial intelligence and technological convergence, expression prediction can be achieved at an early stage with a combination of microarray chip technology.

## 5. Conclusions

In summary, we applied the DNB method to identify a set of biomarkers that change dramatically in the early stages of CRPC and can serve as indicators of the transition of prostate cancer cells from an androgen-dependent state to a castration-resistant state. In addition, the WGCNA method was used to identify core genes, and when combined with DNBs, four genes including SCD, NARS2, ALDH1A1, and NFXL1 were found to be associated with androgen-related signaling pathways in prostate cancer cells. A nomogram model was established by integrating the risk score of the four genes and other clinical characteristics and was verified to reasonably predict the progression-free interval (PFI) of prostate cancer patients treated with androgen deprivation therapy (ADT). High ALDH1A1 and SCD expression were significantly correlated with prostate cancer’s transition from the hormone-sensitive to castration-resistant state. However, since the relevant analyses are based on public data sources (TCGA and GEO), limitations still exist. The expression level of each gene needs to be further validated in different prostate cancer cells. Animal models and more in vivo experiments will be designed to further explore the mechanisms of the biomarkers’ mediation role in CRPC in our future research.

## Figures and Tables

**Figure 1 biomedicines-12-02157-f001:**
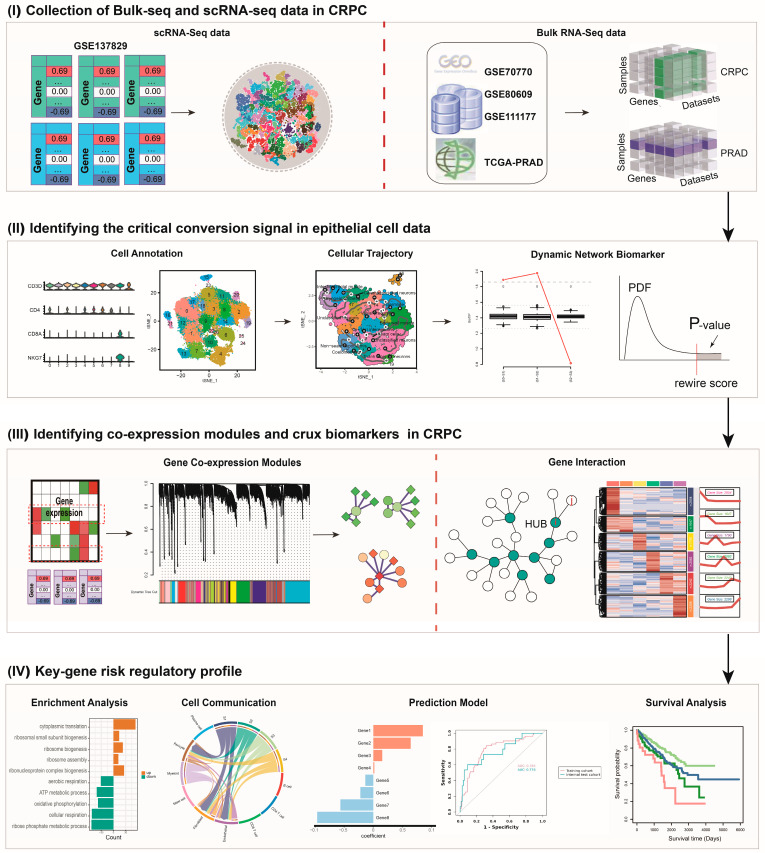
Flowchart of this study.

**Figure 2 biomedicines-12-02157-f002:**
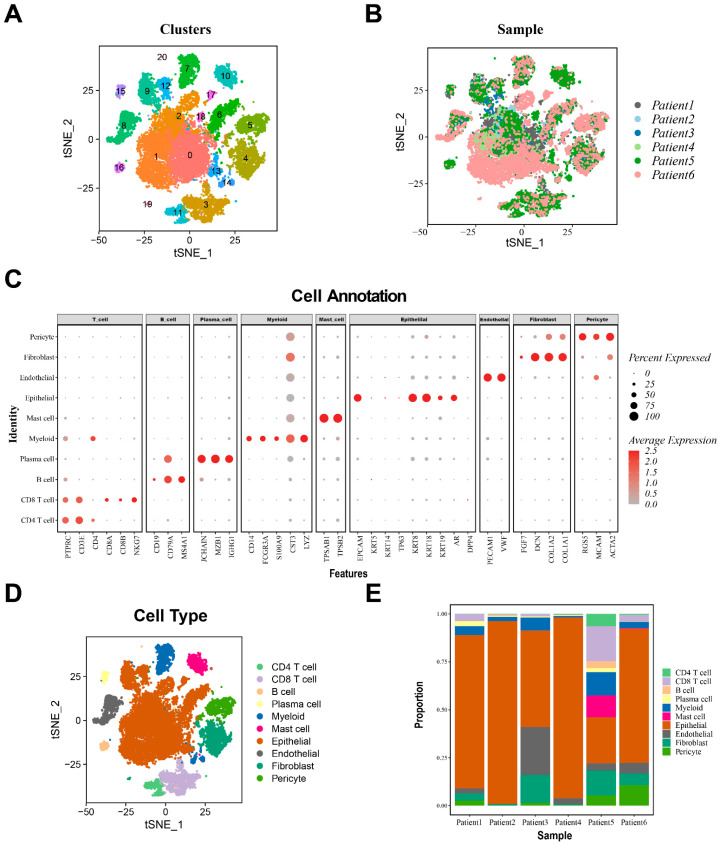
Ten cell clusters with different annotations based on CRPC scRNA-seq data, revealing cellular heterogeneity in CRPC. (**A**,**B**) Dimensionality reduction based on t-SNE algorithm and the distribution of 6 CRPC samples from GSE137829 dataset and 21 clusters were acquired; (**C**) expression level of marker genes in each cell cluster; (**D**) cell cluster annotation based on the composition of marker genes; and (**E**) proportion of different cell types in each sample.

**Figure 3 biomedicines-12-02157-f003:**
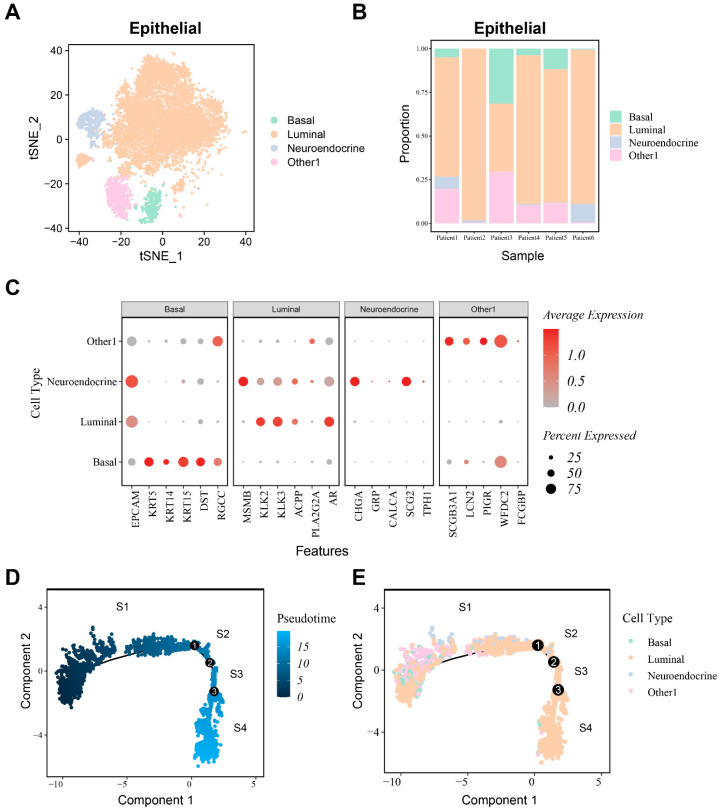
Cell annotation and construction of differentiation trajectories in epithelial cells based on single cell sequence data. (**A**) Annotating epithelial cells according to marker genes; (**B**) proportion of epithelial cell subgroups in different samples; (**C**) marker gene expression level of each epithelial cell subgroup; (**D**) pseudotime differentiation trajectories of epithelial cells; and (**E**) differentiation trajectory states of different subgroups.

**Figure 4 biomedicines-12-02157-f004:**
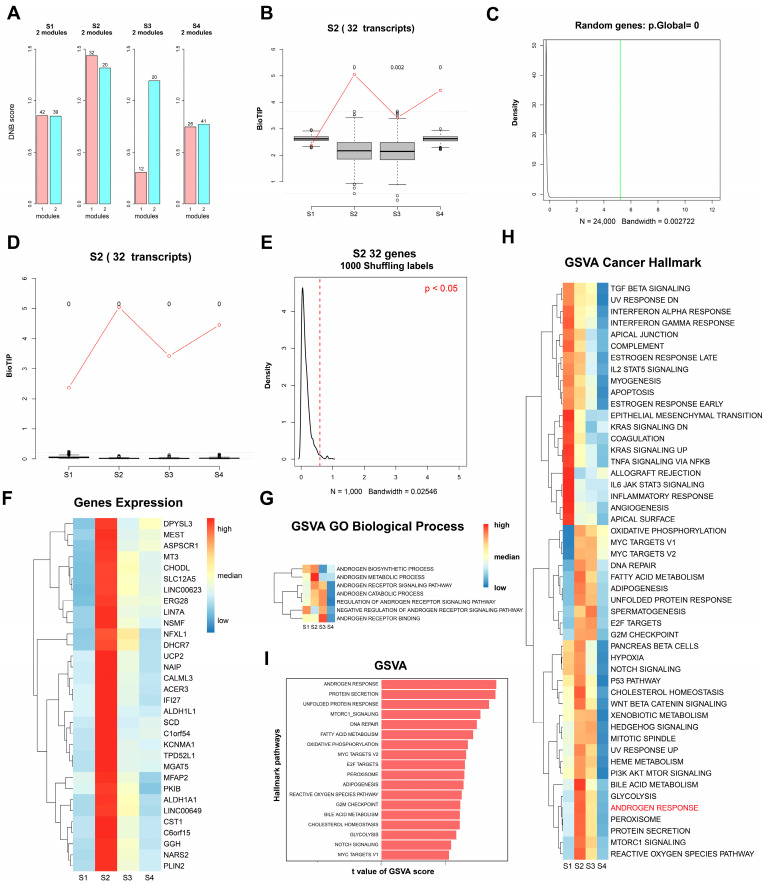
Key transformed cell subgroups and gene modules of epithelial cells. (**A**) Assumed key transforming signal in different cellular differentiation states in epithelial cells. (**B**,**C**) Random score of key transforming signals after shuffling gene labels; (**D**,**E**) random score of key transforming signals after shuffling sample labels; (**F**) expression of key transforming module genes in different epithelial cell states; (**G**) GO BP enrichment state of key transforming module genes in different epithelial cell states; and (**H**,**I**) cancer hallmark enrichment and significance of key transforming module genes in different subgroups of epithelial cells.

**Figure 5 biomedicines-12-02157-f005:**
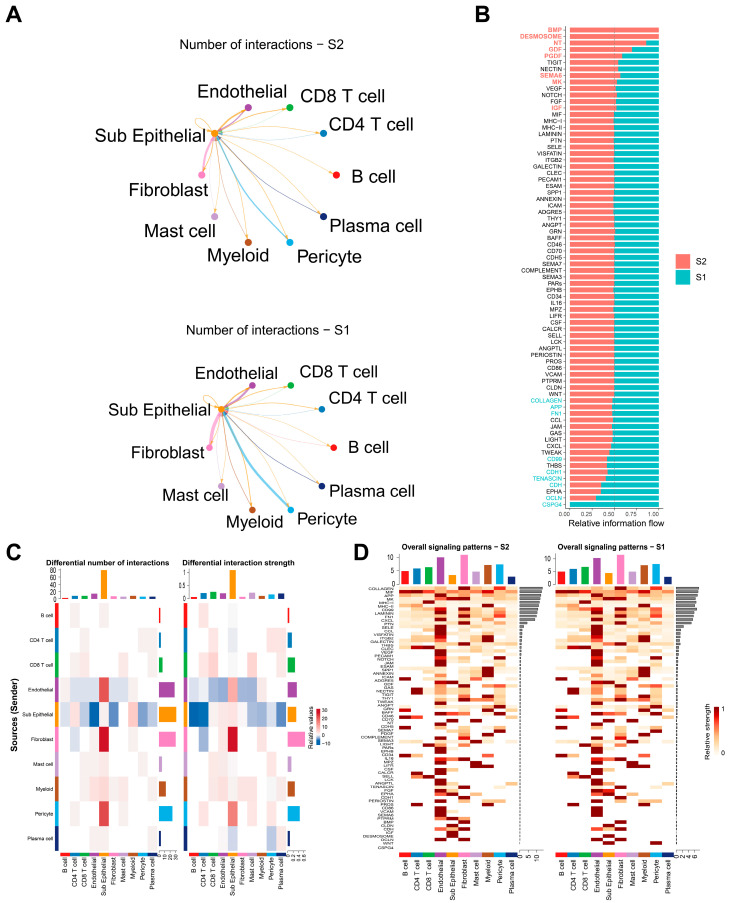
Heterogeneity in cell communication of S1 and S2 epithelial cells in CRPC. (**A**) Cell communication network of S1 and S2 epithelial cells with other cell types (the S2 group is at the top and the S1 group is at the bottom. The size of the spot indicates the number of cells). (**B**) Comparison of the communication strength in different signaling pathways in S1 and S2 epithelial cells. The color “red” on the vertical axis indicates that cell communication was more active in S2 and the color “blue” indicates cell communication was more active in S1. The color “black” indicates that there was no significance between the two groups. (**C**) Cell communication number and strength among different cell types. The color indicates the difference. The color “red” indicates cell communication was more active in S2 and the color “blue” indicates that cell communication was more active in S1. The bar chart on the right indicates that the outgoing signal and the bar chart on the top indicates the incoming signal. (**D**) Heatmap of signaling pathway strength in epithelial cells in the S1 and S2 groups.

**Figure 6 biomedicines-12-02157-f006:**
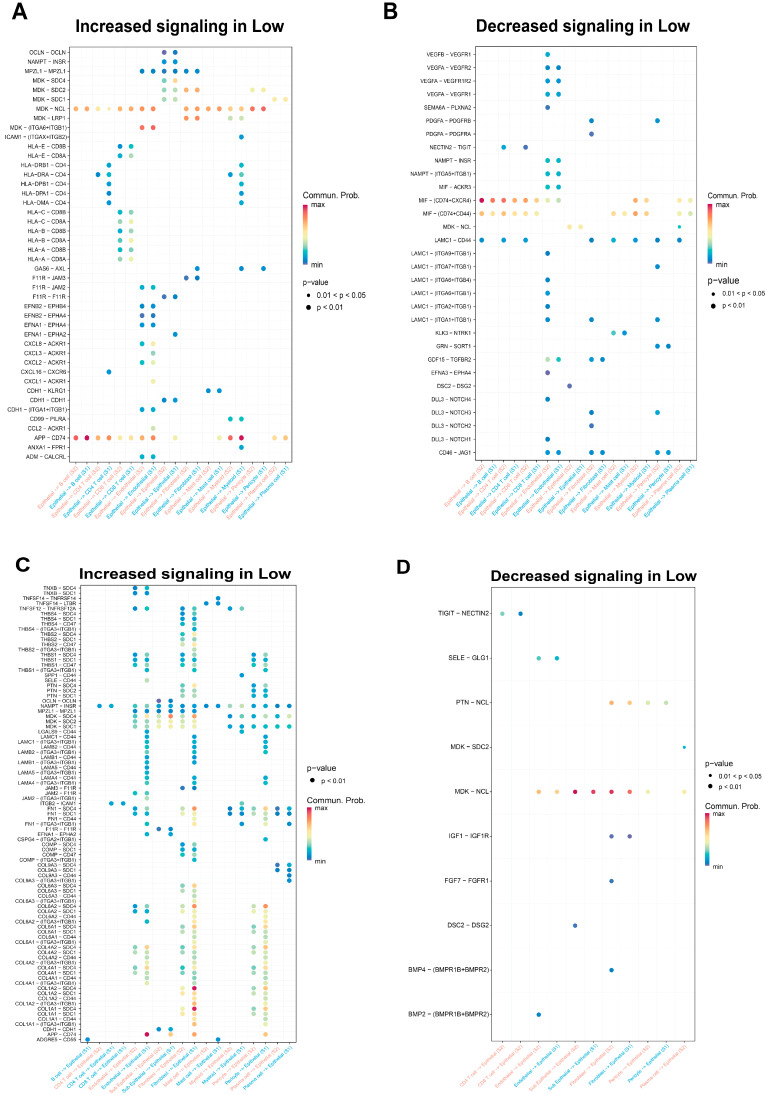
Ligand–receptor differences between S1 and S2 epithelial cells in CRPC. (**A**,**B**) Differences in functional ligand–receptor interactions in epithelial cells in S1 and S2 groups to other cell subgroups. The color “red” on the horizontal axis indicates communication of epithelial cells in the S2 group and the color “cyan” indicates communication of epithelial cells in the S1 group. The color of the spot indicates the cellular communication probability and the size of the spot indicates the significance of the *p* value. (**C**,**D**) Differences in functional ligand–receptors of other cell subgroups to epithelial cells in the S1 and S2 groups. The color “red” on the horizontal axis indicates communication of epithelial cells in the S2 group and color “cyan” indicates communication of epithelial cells in the S1 group. The color of the spot indicates the cellular communication probability and the size of the spot indicates the significance of the *p* value.

**Figure 7 biomedicines-12-02157-f007:**
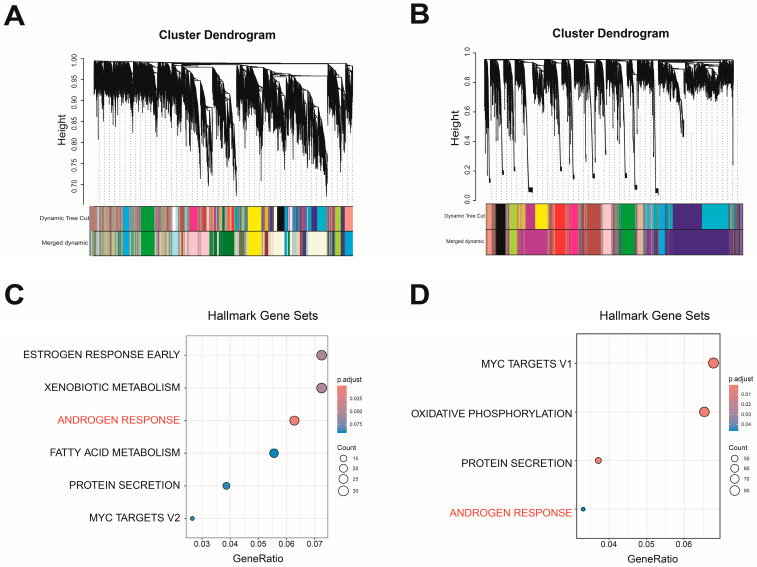
Identification of co-expression modules of androgen-related key genes based on CRPC bulk seq. data. (**A**,**B**) Cluster dendrogram of co-expression network modules in GSE70770 and GSE80609 (1-TOM). (**C**) GO analysis of “midnightblue” co-expressed gene modules in GSE70770. (**D**) GO analysis of “blue” co-expressed gene modules in GSE80609.

**Figure 8 biomedicines-12-02157-f008:**
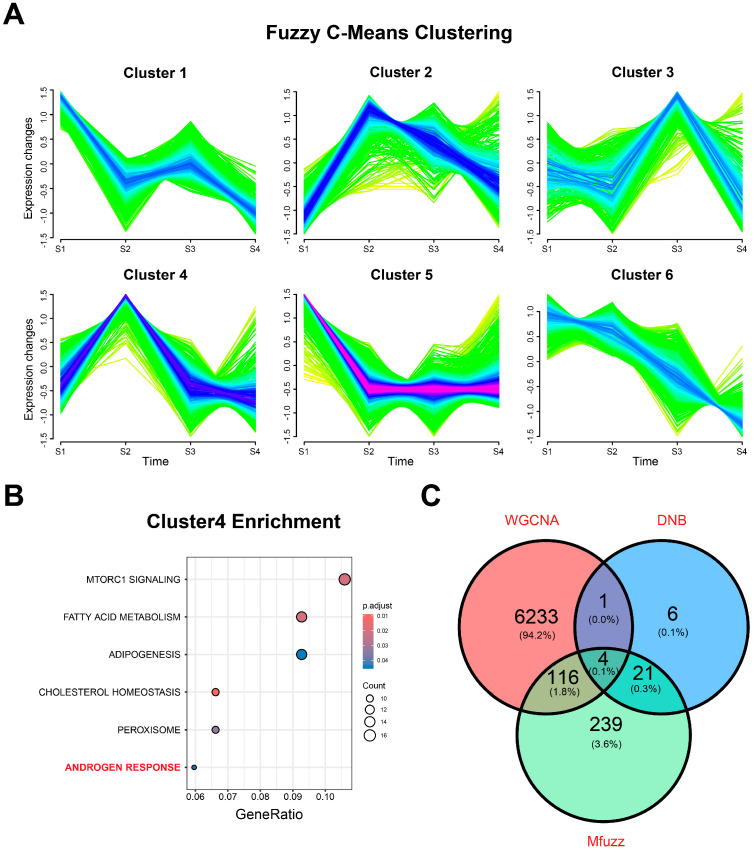
Fuzzy clustering and evaluation of key biomarkers in the key transformation subgroups of epithelial cells. (**A**) Using Mfuzz to clarify the dynamic change in neighboring genes in the key transforming signaling modules in different states of cytodifferentiation in epithelial cells. (**B**) Hallmark enrichment analysis of the cluster 4 gene set. (**C**) Intersection Venn plot of genes in the WGCNA, DNB, and soft clustering analysis.

**Figure 9 biomedicines-12-02157-f009:**
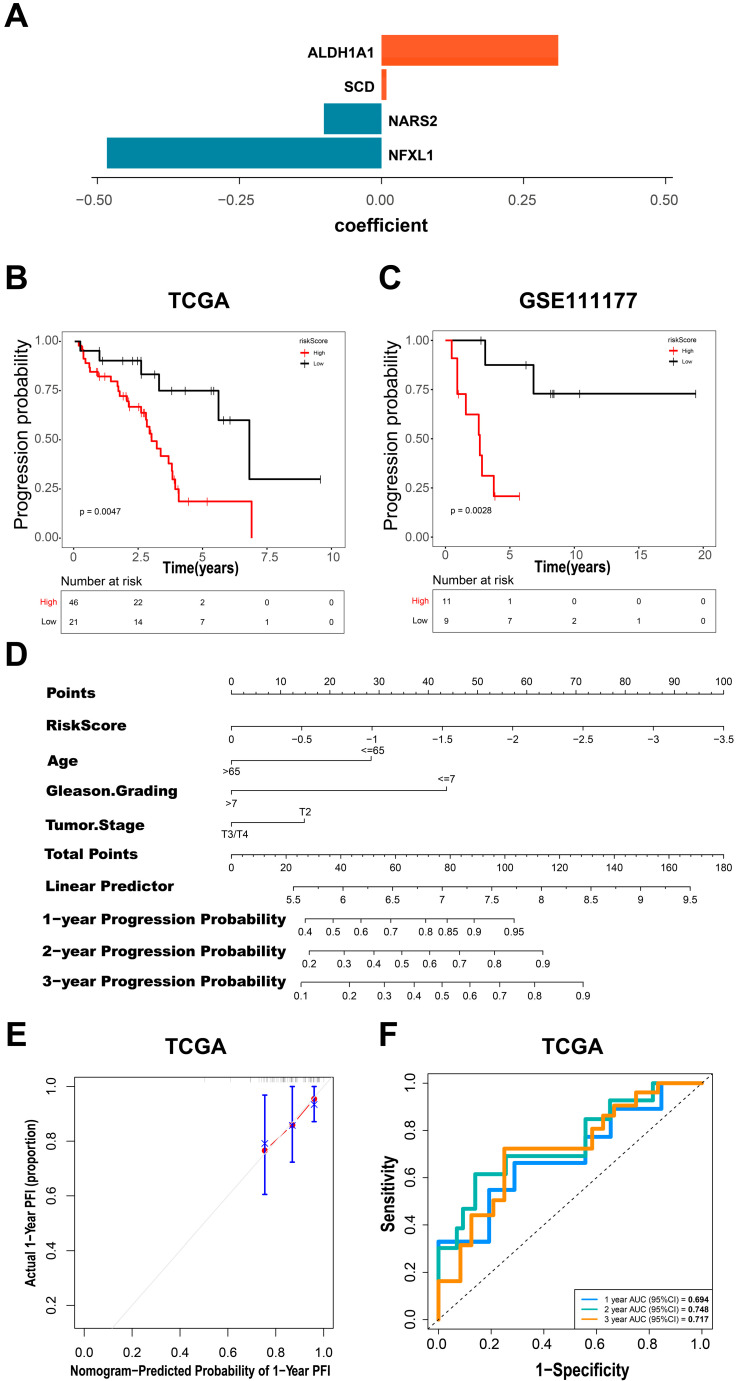
Key transformed cell subgroups and gene modules of epithelial cells. (**A**) Risk score of the 4 key biomarkers. (**B**) Kaplan–Meier curve of the TCGA training cohort. (**C**) Kaplan–Meier curve of the GSE111177 validation cohort. (**D**) A nomogram combining the risk score, age, Gleason grade, and tumor stage was developed to predict the 1-, 2-, and 3-year PFIs of patients who underwent ADT in the TCGA cohort. (**E**) A 1-year calibration analysis of the TCGA cohort nomogram. (**F**) ROC curves of multiple time points (1 year, 2 years, 3 years) of the PFI in the TCGA cohort.

**Figure 10 biomedicines-12-02157-f010:**
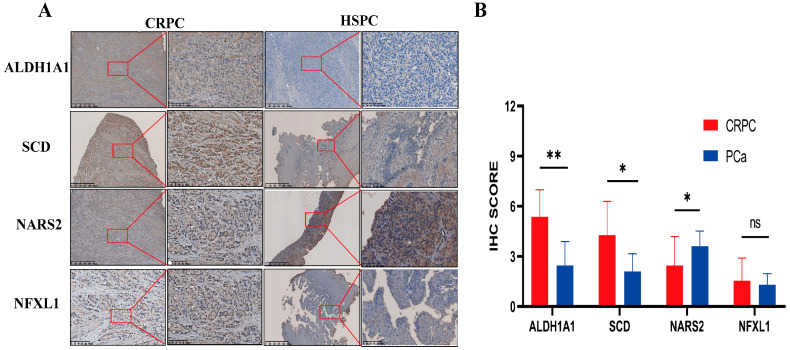
Representative photo images and histograms of ALDH1A1, SCD, NARS2, or NFXL1 protein expression levels in CRPC or HSPC sample tissues. (**A**) Representative IHC images showing high ALDH1A1 and SCD expression in CRPC tissue and high NARS2 expression in HSPC tissue. (**B**) Histograms of ALDH1A1, SCD, NARS2, and NFXL1 expression levels. Scale bars, 625 μm and 100 μm. * *p* < 0.05, ** *p* < 0.01; ns, not significant.

## Data Availability

Publicly available datasets were analyzed in this study. These data can be found here: https://www.ncbi.nlm.nih.gov/geo/query/acc.cgi?acc=GSE137829, accessed on 1 September 2020; https://www.ncbi.nlm.nih.gov/geo/query/acc.cgi?acc=GSE70770, accessed on 30 July 2015; https://www.ncbi.nlm.nih.gov/geo/query/acc.cgi?acc=GSE80609, accessed on 22 April 2018; https://www.ncbi.nlm.nih.gov/geo/query/acc.cgi?acc=GSE111177, accessed on 11 October 2018; https://portal.gdc.cancer.gov/projects/TCGA-PRAD, accessed on 16 December 2019.

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
