# Peer review of "Identifying Key Genes as Progression Indicators of Prostate Cancer with Castration Resistance Based on Dynamic Network Biomarker Algorithm and Weighted Gene Correlation Network Analysis"

_biomedicines, 2024, doi:10.3390/biomedicines12092157_

Round 1

Reviewer 1 Report (Previous Reviewer 3)

Comments and Suggestions for Authors

Dear Authors,

Biomedicines-3163950

Identifying key genes as indicators of progression of prostate cancer with castration resistance based on dynamic network biomarker algorithm and weighted gene correlation network analysis by Liu et al identified a risk associated with key genes SCD, NARS2, ALDH1A1, and NFXL1 in CRPC. This analysis is very interesting. However; this study needs further investigation.

Major Comments:

1.     Are these SCD, NARS2, ALDH1A1, and NFXL1 are AR target genes.

2.     Is there any correlation between the Gleason score and these 4-key genes

3.     It is nice to see the validation of these genes in primary and CRPC (LNCaP, 22RV1, C4-2) cancer cells.

4.     These analyses are simple predictions and further validation is required.

Comments on the Quality of English Language

English Edit needed

Author Response

Reviewer 2 Report (New Reviewer)

Comments and Suggestions for Authors

This paper explores the gene set associated with prostate cancer progression using a combination of WGCNA and DNB. It is not particularly novel, just a combination of a number of existing methods. The application is also inadequate in many respects, and there is no sufficient validation, etc., as is done in similar papers. In addition, the method of describing the matemeso is also insufficient, making it difficult for any journal to accept the paper as a scientific paper.

Figure 1 is easy to understand with plenty of figures, but in general, it is inappropriate as a figure for a research paper and should be moved to the Supplementary section. Rather, the data analysis should be presented at a level of information that can be reproduced in the data analysis.

The structure of the materialeso needs to be modified significantly. For example, the Supplementary should include some evidence of the excluded data, such as a histogram of the data from which outliers were selected to be excluded. Overall, the description of the methodological philosophy and design is weak, simply listing the tools used and their options. The first half of the material should describe the generalized contents so that the reader can understand the overall picture of data processing and the authors' philosophy. It would be easier to read if the second half of the document contains implementation details such as options for individual processes.

For example, when the risk score in 2.10 is calculated from multiple genes, on what basis is the number of genes determined? Does the inclusion of a large number of genes cause multicolinearity? Various questions remain, such as

The conclusion is that four biomarkers were found, but the biochemical significance of these remains unclear. Although the discussion enumerates functions reported in other papers, it is necessary to first discuss the novelty of this method and its advantages, which are not the essence of this paper.

The nomograms are not validated, and the paper simply includes the results of supervised learning, which is not sufficient for a research paper.

Author Response

Reviewer 3 Report (New Reviewer)

Comments and Suggestions for Authors

The manuscript uses at least 30 different algorithms to identify genes that may or may serve as diagnostic markers, and possibly predict patient outcomes in PrCa. Hundreds of publications identify a certain gene set or marker pane with alleged prognostic/diagnostic value, but none of them have made any differences in patient stratification and personalized medicine, or therapy planning. Therefore, it is important to clarify where is the novelty of this study. In my opinion, it is - if anything - related to the combination of bioinformatic analyses and algorithms, and in combining bulk and single-cell RNA data for a more integrated analysis. Also, the developmental trajectories introduced here will be of interest to other researchers. For this, it makes sense to re-analyze the identity of epithelial cells in PrCa biopsies, which may also be tf interest for some readers, as this is a matter of debate. The classification into 4 stages based on developmental trajectories is quite interesting and I haven't seen such data before, but also this part suffers quite a bit from the general shortcoming of this manuscript: the lack of validation that is NOT based on analysis of mRNA data. This is the main bottleneck for the manuscript to be truly informative: the findings are difficult to translate into true meaning. 

One of the main findings of the DNB analysis is that AR itself represents a core gene for prostate epithelial differentiation and this is of course not a very novel finding. In many analyses AR has popped up as the most important, or even the only statistically relevant gene that distinguished various stages of PrCa progression and differentiation. This is also apparent in the data shown in Fig. 7 and the matching paragraph in the text: AR has written its signature over everything. Not exactly a new finding... 

SImilar issues apply for the identification of "differentially expressed genes" between the S2 and other stages of epithelial differentiation. 

Concerning the "cell communication" analysis shown in Fig. 5, and the ligand/receptor interactions shown in Fig. 6 - I really wonder - isn't this all theoretical? Which ligand/receptor, or cell-cell interactions, for example, can be identified solely from mRNA expression data? The concept is appealing, but I think the mRNA data are so convoluted, and distant from biochemical interactions, that it is notoriously difficult, if not impossible, to draw any conclusions about the "cell communication" that's going on inside tumor biopsies. At least not without any functional verification(s). This part is entirely hypothetical and borders on speculation, I believe the authors should clarify what makes them so confident about this and also consider picking an exemplary interaction and validating it... this will be rather difficult, I believe.  In my opinion, such findings may or may not be interesting and relevant - we will never know without doing experiments. 

The risk stratification shown in Fig. 9 would potentially benefit from looking at additional data for these 4 genes, e.g., adding AR into the "big picture" as this has been demonstrated as critical for almost all steps in PrCa development and progression. Does AR correlate with any of the findings shown in Fig. 9? That would add an interesting dimension. 

Thus, I think its mainly on the methodological side that makes this manuscript interesting, while the "candidate genes" identified at the end of this study, or the interactions etc. arent very likely to make any difference for patients.  

It appears that some independent validation NOT based on mRNA expression, and NOT identified by purely bioinformatic methods, the authors were asked in a previous round of review to add some Western blot analyses. 

The cell lines used here appear both representatives for castration-resistant prostate cancer.  I cannot really identify a substantial difference between their response to androgens.

The 22rV1 cells are derived from the  CWR22 xenograft, and generated by castration-induced regression and relapse in the mouse model. C-42 is in fact a castration-resistant subclone of LNCAP cells, also generated in vivo from castrated mice - meaning, it is castration-resistant by definition just as the 22rV1 cells and even generated by the same methods. 

Even parental LNCaP cells have mutations in both alleles of the androgen receptor (AR) that provide them with hypersensitivity to very low levels of androgens (or other steroids) ... this is already an adaptation to androgen replacement therapy. 

So the "differences" observed in Figure 10 are marginal and, in my opinion, represent baseline differences in expression levels of 2 genes between 2 cell lines, likely without much meaning of consequences. There cannot be any functional conclusions made from the small and non-significant differences that are also not known to be of any functional relevance. 

Two possibilities would exist: 1) the authors check if the knock-down or knock-out of any of these genes may have any functional consequences for cell growth, AR response, drug sensitivity, etc. 

2) the authors check additional data bases for correlation of these genes with patient outcome and survival, maybe also based on protein expression (like proteinatlas), and see if they can get more conclusive clues from there. 

SMALL ISSUES

I wonder why there are sections labelled in yellow? I assume this is a resubmission and these section were added in a previous round of reviews to address the reviewers question. Here, however, this is somewhat distracting. 

the heading for the chapter 2.5. Gene Set Enrichment Analysis (GSVA) contains a typo - obviously, it should be called 2.5. Gene Set Variation Analysis (GSVA). 

Explanation of the DNB method is very complex and difficult to follow, or comprehend. Maybe the authors can provide a schematic representation of all of these steps that were taken subsequently and add it to the manuscript, maybe supplemental is sufficient. 

Comments on the Quality of English Language

the english language use is okay, no major issues 

Round 2

Reviewer 1 Report (Previous Reviewer 3)

Comments and Suggestions for Authors

Thank you.

Reviewer 2 Report (New Reviewer)

Comments and Suggestions for Authors

The authors revised the manuscript to insist on the uniqueness of their own study. Also, they added validation data. The current manuscript is enought to be published.

This manuscript is a resubmission of an earlier submission. The following is a list of the peer review reports and author responses from that submission.

Round 1

Reviewer 1 Report

Comments and Suggestions for Authors

1) General comments

Liu et al. performed data mining and an in silico analysis of datasets from the Cancer Genome Atlas (TCGA) and Gene Expression Omnibus (GEO) and a few other publicly available databases, using bioinformatic tools commonly available on the internet. They combined Dynamic network biomarker (DNB) and weighted gene co-expression network analysis (WGCNA) and selected four genes to construct a risk prediction model for “progression” of castration-resistant prostate cancer (CRPC).  Although the research topic is intriguing, there are major flaws in data presentation and interpretation; some of these are detailed below. 

2) Specific comments of revision

1.  As described in page 4, line 7, single-cell RNA sequencing data from GSE137829 (www.ncbi.nlm.nih.gov/geo/query/acc.cgi?acc=GSE137829) includes neuroendocrine prostate cancer (NEPC) which is now recognized as a distinct biological and clinicopathological entity, treatment-related NEPC (tNEPC).  However, pseudotime trajectory analysis in the present study revealed that only a few proportions of neuroendocrine cells were included in tumor cells of these cases. Reasonable explanation is necessary for this discrepancy of the data to ensure analytical accuracy.

2. Because of its importance, more detailed method for the selection of four gene (SCD, NARS2, ALDH1A1, and NFXL1) should be described in the manuscript.

3. Please clearly state case-selection criteria for CRPCs in dataset of GSE70770, GSE80609, and GSE111177.

4. Definition of “disease progression” of CRPCs in the present study is need to be described. Were all cases non-metastatic CRPCs? In addition, if possible, survival data should be added.

5.  Higher resolution files are necessary for Figures 5 and 8, which are very hard to figure out in the present form.

6. As stated the last part of the manuscript, the authors did not perform in vivo analysis using independent cohort of CRPC to validate their risk predicting models.  This limitation should be more precisely described in discussion, because it is mandatory for this kind of studies.

Comments on the Quality of English Language

Minor editing of English language required including proper use of abbreviations.

Reviewer 2 Report

Comments and Suggestions for Authors Finding key genes as progression indicators is of great importance. The authors did some analysis using a DNB method to identify the key genes and also developed a risk prediction model with these key genes. My comments to improve the manuscript:   Major: 1. Page 4, session 2.1, all functions mean? Where do the “FindNeighbors”, “FindClusters” functions come from? 2. Page 7, sesision 3.1, 23,987 cells from 6 samples? How many cells from each sample? From Figure 2B, it seems patient 6 has a lot of more cells than other patients? 3. Page 4, which package did the authors use to identify DEGs for bulk RNA-seq? 4. Improve the language 5. Use the full names for  all the abbreviations when they appear the first time such as MFUZZ (session 2.9), CM (session 2.1, page 4), NEPC (session 2.1, page 4) 6. Page 9, how was the developmental trajectory constructed? 7. Page 11, I am not sure if I understand section 3.2. The authors claimed that the DNB module of 32 genes have a greater expression level than genes in other epithelial cell subgroups (Figure 4F). From what I can see in Figure 4F, it’s state S2 with greater expression than other states. I don’t know what other epithelial cell subgroups refer here. 8.Figure 4G, 4H, and 4I, different states have different patterns. Did the authors make conclusions based on state S2 for section 3.2? 9. It’s very difficult to see the texts in Figure 4 and Figure 5, Figure 6A and 6B, Figure 8.   Minor: Page 6, session 2.10, it should be “gene Expi” refers to the gene expression level. Not “I” refers to the gene expression level. Comments on the Quality of English Language

Quite poor English in many places. Some examples:

1. In the abstract, "By combining single-cell RNA sequence and bulk RNA sequence data in we identified a set of DNBs", bulk RNA sequence data in what?

2. page 11, "it was discovered that all genes in the module have a greater expression level than did genes in other epithelial cell subgroups (Figure 4F)."

Reviewer 3 Report

Comments and Suggestions for Authors

Dear authors,

Identifying key genes as progression indicators of prostate cancer with castration resistance based on dynamic network biomarker algorithm and weighted gene correlation network analysis by Liu et al demonstrate the Dynamic network biomarker (DNB) method and weighted gene co-expression network analysis (WGCNA) to identify key genes associated with progression to castration-resistant state in prostate cancer via the integration of single-cell and bulk RNA sequencing data. Based on the gene expression profiles of CRPC in the GEO dataset, the DNB method was used to clarify the condition of epithelial cells and find out the most significant transition signal DNB modules and genes included. Authors calculated gene modules associated with the clinical phenotype stage based on WGCNA. The results of nomograms, calibration plots, and ROC curves confirmed the good prognostic accuracy of risk prediction model. By combining single-cell RNA sequence and bulk RNA sequence data in we identified a set of DNBs, whose roles involved in androgen-associated activities indicated the signals of prostate cancer cell transition from androgen dependent state to a castration-resistant state. In addition, a risk prediction model including the risk score of 4 key genes (SCD, NARS2, ALDH1A1, and NFXL1) and other clinical pathological characteristics was constructed and verified to be able to reasonably predict the prognosis of patients receiving ADT. In summary, 4 key genes from DNBs were identified as potential diagnostic markers for patients treated with ADT and a risk score-based nomogram will facilitate precise prognosis prediction and individualized therapeutic interventions of CRPC. This is an interesting study; however; it needs additional data to make the manuscript stronger.

Major Comments:

  1. Authors could validate these four genes SCD, NARS2, ALDH1A1, and NFXL1 in CRPC patients’ samples at least by PCR.
  2. All the 4-genes are transcript level; however, these need to be validated in CRPC (C4-2 ) cell lines with antibody
  3. Please take care of iThenticate report (35%) should be less.